# Historical Ecology: A Robust Bridge between Archaeology and Ecology

Carole L. Crumley 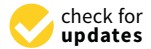

Department of Anthropology, University of North Carolina, Chapel Hill, NC 27599, USA; crumley@live.unc.edu

**Abstract:** How can the disintegration of ecosystems, the foundation of life on Earth, be halted and these critical systems be rehabilitated? For scholars, the action list is long: increase the pool of expertise by engaging all relevant knowledge communities, collect rapidly disappearing data, analyze with both familiar and new methods, and apply the results of actionable science to policy and practice. This enormously complex and urgent activity requires an integrated research framework with the flexibility to accommodate the global diversity of places, peoples, and processes and to examine future options. Based on evidence of environmental change and human activity, the framework termed historical ecology assembles tools to construct an evidence-validated, open-ended narrative of the evolution and transformation of specific ecosystems and landscapes. Welcoming knowledge from scholars and communities of both heritage and practice, this comprehensive and systemic understanding offers insights, models, and ideas for the durable future of contemporary landscapes. The article evaluates how practitioners could adjust aspects of practice and improve access to policy makers, and the discussion applies to regions and localities everywhere.

**Keywords:** historical ecology; regions; archaeology; history; ecology

## 1. Finding Tools to Meet the Future

Multiple crises, which menace not just humanity but all life on Earth, are unfolding. With its link to global warming, heedless management is accelerating the collapse of ecosystems everywhere; this means that all practitioners, whether they are scholars or anchored in a landscape, must collaborate to meet this unprecedented challenge.

How can the disintegration of ecosystems, the foundation of life on Earth, be halted and these critical systems be rehabilitated? For scholars, the action list is long: increase the pool of expertise by engaging all relevant knowledge communities, collect rapidly disappearing data, analyze with both familiar and new methods, and apply the results of actionable science to policy and practice [1,2]. This enormously complex and urgent activity requires an integrated research framework with the flexibility to accommodate the global diversity of places, peoples, and processes and to examine future options.

Based on evidence of environmental change and human activity, the framework termed historical ecology assembles tools to construct an evidence-validated, open-ended narrative of the evolution and transformation of specific ecosystems and landscapes. The term historical ecology includes humans as a component of ecosystems' evolution and defines history in a way that goes beyond the written record to encompass both the history of the Earth system and the social and physical past of humans and other species. The core idea of historical ecology is that all sources of knowledge are combined to understand perspectives on the past in a specific place, so that its future can be more wisely managed [3–5].

Welcoming knowledge from scholars and communities of both heritage and practice, this comprehensive and systemic understanding offers insights, models, and ideas for the durable future of contemporary landscapes. In this article I examine historical ecological approaches to landscapes but also to oceans, seas, rivers, lakes, marshes, springs, and other moist areas, which I collectively term waterscapes. I evaluate how practitioners could

adjust aspects of practice and improve access to policy makers. While examples herein rely on my greater familiarity with European and North American research, the discussion applies to regions and localities everywhere.

## 2. Historical Ecology in Disciplinary Contexts

Among several methodological and theoretical approaches that detail and track key elements of the human–environmental nexus and the linkages of biotic and abiotic agents and their behaviors through time (e.g., resilience, ecosystem dynamics, environmental history, and landscape biography models), the framework of historical ecology offers a comprehensive and integrated reach across knowledge sectors and clear strategies for social justice, collaboration, and application [6,7].

There is no particular need to identify one's work as historical ecology, as several other approaches employ similar principles [8]. However, the term is widely familiar, teasing ecology to embrace the historical sciences and history to learn ecology. Historical ecology is an umbrella term describing multi-faceted research programs that assure researchers and stakeholders the advantages of diverse perspectives, the means to evaluate and share information, and a community of practice. Historical ecology unites a group of core disciplines—archaeology, anthropology, ecology, geography, and history—and draws on parallel developments in these fields.

In archaeology, historical ecology derives, for the most part, from the standard practice in landscape and environmental archaeology, which routinely amalgamates information from disparate sources about the past of sites, places, landscapes, and regions; the focus is to explain human management approaches to their environments and to evaluate their consequences [9]. In ecological anthropology, historical ecology emerged through frustration with an earlier cultural ecology approach, which could not accommodate time or change [3].

In geography, early perspectives on landscapes [10,11] and temporality [12,13] reflect an enduring association between the disciplines of anthropology and geography. The holistic approach of geographer Karl Butzer [14–17] has influenced generations of practitioners in several fields of study. Along with the perspectives of naturalists [18] and environmental historians [19–22], historical ecology unites disciplines in both theory and practice.

In ecology, management approaches derived from forest history, conservation biology, and restoration ecology are also termed historical ecology or applied historical ecology [23–27]. Historical ecology is widely used as a framework to co-manage cultural heritage and natural resources and to engage in restoration and conservation ecology. For instance, the Society for Ecological Restoration International [28] structures its research and instructional programmes around historical ecology. For SERI, restoration embraces the nature/culture relation, engages all sectors of society, and enables full and effective participation of local, indigenous, and disenfranchised (LID) communities. This wide utility underscores the importance of integrating the historical sciences with ecology in their several forms and at multiple scales, while confirming a key connection between ethics and practice.

There is considerable overlap in strategy, tools, and techniques. For example, in the study of vanished landscapes, paleoecology and archaeology track changes over time in plant community dynamics, soil development, sediment history, climate, and hydrology; these are only some of the topics common to both fields [29,30]. Such close collaborations have helped archaeologists and paleoecologists to reconstruct a remarkable span of landscape history, from the ancient landscapes of early hominids [31] to early agrarian landscapes [32–34] and historic gardens [35,36].

## 3. Climate Change Remodels Landscapes

The availability of water will be an increasingly constraining variable in finding regions suitable for growing food as well as fibre, fuel, and fodder [37]. An ancient example comes from North and West Africa, which—due to monsoon rains in the late

Pleistocene—was a region of springs and permanent lakes; the population subsisted on abundant resources that supported hunting, fishing, and gathering. With settlements anchored near water and easy access to diverse biomes and ecotones, they voyaged like ancient mariners across arid and desertic areas to find a large selection of food in water bodies, woodlands, savannahs, and oases [38]. Drawings in rock shelters and caves depict their dead as swimmers in the sand, between the islands of life and the afterlife [39]. After ca. 6000–5000 BP, the region became steadily drier; the population began to practice pastoralism and moved to the more reliable water source of the Nile River, bringing their religion with them, and founding one of the great civilizations of the world [40].

Much can be learned from this example. The proto-Egyptians were masters of ecotonal resource use which, for a long time, enabled them to avoid moving elsewhere. While the move from a region of diminishing availability of water to one of abundance took centuries, today we do not have the luxury of time to modify our practices and adjust to new conditions; we must make the necessary changes within years and decades. This will require the gift of flexibility: the ability to conceive of novel solutions but also to draw upon the diversity of solutions that humanity has applied. That will require deep knowledge of the species, habitats, ecosystems, and ecotones upon which humans rely and a sense of reverence and protection for the sources of our food and water.

It is once again time to re-investigate how the human impact on the planet can be lessened. The relatively easy choices (e.g., recycling, less polluting materials) have been made, but more difficult choices that require the radical renovation or the abandonment of both brick-and-mortar infrastructure and supply networks have yet to be undertaken. This reinvention of the built environment and networks of commodity management must be aligned once again with regional ecology and climate [41]. A good place to begin is to coordinate ongoing research at regional and territorial scales to deliver an integrated approach to those who will plan the future. While the survey below is geographically limited, it can point to some areas worthy of greater attention.

**4. Landscapes: Building Frameworks and Standardizing Practice**

Persistent landscape types (forests, arable land, wetlands) and functions (community-managed land, sacred places) are of particular interest because considerable evidence can deepen the baselines for key resources and activities. The research designs of landscape ecologists and archaeologists can easily accommodate other fields of study (e.g., heritage and regional planning; climate change; sustainable management).

Early work in historical ecology focused primarily on landscape types. Many established research groups study a particular mountain, forest, or grassland landscape: an early example of historical ecology as policy is the U.S. Geological Survey's work in the southwestern Rockies [42]. An early national approach is the Swiss Federal Institute for Forest, Snow and Landscape Research [43], founded in 1885 and using historical ecology since the 1990s; researcher-driven transdisciplinary work focused on the Pyrenees began about the same time [44].

While scaling, politics, and other issues impede more recent global-scale management, mountains, uplands, and forests are often studied at regional and trans-border scales [45–52]. Landscapes that were once managed as commons have been brought back into view, using a historical ecology approach termed 'environing' [53]. Scholars and practitioners under the aegis of the International Association for the Study of the Commons has planned a forest commons conference [54].

The traditional and sustainable management practices of LID communities have been explained and promoted in several contexts; good examples are several decades of regional and cross-boundary work to explain Saami practices to Scandinavian governments [55] and to rehabilitate traditional solutions [56,57]. Connections between the ongoing disappearance of African wetlands and the expansion of agriculture offer another example of how the regional study of shifting relations among landscape elements can signal major issues such as the decline of biodiversity or looming water shortages [58–60]. Funded

by the European Research Council [61], the MEMOLA project studied four mountainous European landscapes (in Spain, Albania, and Italy) to analyze agroecosystems that both maintain tradition and ensure the livelihood of rural communities over time.

These place- and region-based landscape projects serve the historical record, guide current decisions, and strengthen future management. Publishing outlets for local and regional work are expanding, notably the interdisciplinary journal Regional Environmental Change [62], the goal of which is to understand change, causation, and impacts at all territorial scales between the local and the global, whether they are defined by natural criteria (e.g., watersheds, ecosystems) or by human activities (urban areas/hinterlands).

Among the newest of international programs in this arena is the UNESCO BRIDGES global research coalition. BRIDGES [63] aims to integrate with UNESCO's Management of Social Transformations [64] intergovernmental science program. The aim of the coalition is to better integrate humanities, social science, and local and traditional knowledge perspectives into research, education, and action for global sustainability at local and territorial scales.

The European Research Council has funded additional future-oriented landscape projects. HERCULES [65] has a focus on the empowerment of public and private actors to protect, manage, and plan for sustainable landscapes of significant cultural, historical, and archaeological value at local, national, and pan-European scales. The European Commission has funded TERRANOVA [66], which trains next-generation researchers by charting shifting energy regimes as they have impacted land use strategies in Europe and demonstrates how landscape managers can draw on place-based solutions. The Commission also funds HERILAND [67], which addresses heritage management by exploring new ideas, tools, and training to ensure that interdisciplinary, research-based heritage, landscape management, and spatial planning are positively integrated with business activity, development, and democratic decision making.

At the national level, the U.S. National Park Service [68] published their strategy to manage cultural resources and climate change [69]. This inspired a group of researchers to form Climate Change Strategies and Archaeological Resources [70]. The group wishes to enhance archaeology's effectiveness with policy makers to increase knowledge about the multiple challenges that climate change has posed to the valuable and irreplaceable historical record.

Researcher-led coalitions have established the Historical Landscape Ecology Working Group [71], where members of the International Association for Landscape Ecology [72] and the International Association of Landscape Archaeology [73] share research and perspectives [74,75]. A group of researchers from many disciplines formed the project Integrated History and Future of People on Earth [76] in 2004. Founded on the principles of historical ecology, this global network has projects that feature collaboration with LID communities.

Global warming has already begun to transform familiar landscapes in ways that are difficult to predict in detail. French growers have begun to prepare for the future of their storied wines by working closely with climatologists, biologists, economists, sociologists, geographers, and geneticists to begin the process of adapting to anticipated climate change [77,78]. In Sicily, olive growers have embraced historical ecology to prepare for changing conditions [79]. A geological and archaeobotanical approach to land-use change is used to identify and protect High Nature Value (HNV) Sicilian farmlands for the future [32,80].

## 5. The Land-Water Ecotone: Policy-Oriented Research Design

Seventy-one percent of the Earth's surface is water. Oceans and brackish water comprise about 97% of that quantity and play an important role in feeding the world's population. Fresh water—from rivers, springs, streams, lakes, and ponds—while accounting for only 2.5% of the planet's water, is vitally important for human consumption and for agriculture [37,81,82]. Throughout human history and just like the early Egyptians, people have chosen to live at ecotones, where several ecosystems converge and the biotic diversity

is greatest. Lacustrine and riverine environments nourished our species; especially favored were places where a short distance separates fresh and saltwater, where rivers meet the sea.

Archaeologists can read past and present landscapes that allow them to find these places, even as shorelines and the courses of rivers have remodeled the landscape. There, they excavate the debris from long-ago expeditions to locate food, entwining human activity with the health and behavior of many species. Thus, sites containing the remains of kills and catches enter the archaeological record and allow deep knowledge of both prey and their environments. Particularly important in wildlife introductions and species history, genetic analysis can be undertaken with material from archaeological sites and collections [83]. These sites are time capsules for land and water species' histories, ecosystems, water quality, and the resource management and culinary practices of the searchers. As millennia-deep ice sheets melt, rising sea and lake levels and floods threaten these shoreline archives [84,85].

An important advantage of collaboration among archaeologists, zooarchaeologists, and paleoecologists is that baselines—a guideline or beginning point of reference—chart the history of entities (such as species) or phenomena (such as salinity) over time [86–88]. Thanks to a variety of techniques, it is now possible to trace species and ecosystems over centuries and millennia, enabling the assessment of shifts in climate, ecosystems, and species' abundance and health; these tools are especially useful in conservation [89–91].

Marine historical ecology (MHE) offers fruitful applications of historical ecology and environmental history to marine ecosystems [92], shorelines [93–96], and island ecosystems [97–99]. The continuing importance of coastal wetlands is underscored in the long-term analysis of their storm protection [100]. Much of this work has been accomplished by collaborating regional groups. The vibrant alliance of scholars with indigenous groups along the North Pacific façade has re-invigorated ancient practices to ensure the health of coastal resources such as herring and clams [101,102]. In south Florida, a deep-time study of marine resources in the Gulf of Mexico traces more than a thousand years of fishing and collecting [103]. The Distributed Long-term Observing Networks of the Past [84] assess human behavior and environmental change in the Arctic and subarctic regions over space and time. All these collaborations engage multiple knowledge communities, both of heritage and of practice, while addressing climate change [104–108].

A seminal article [109] outlines how MHE researchers have taken the integrated methods of historical ecology directly to policy makers, and then followed up by analyzing whether the desired effects are reached and maintained. This muscular approach is a blueprint for action, precisely what is needed to ensure that historical ecology changes the thinking of policy makers and is thereby standardized.

The article identifies six policy themes: climate change, biodiversity conservation, ecosystem structure and function, habitat and seabed integrity, food security, and the importance of including social and economic considerations and facilitating 'bottom-up' governance to balance 'top-down' policies. It would not be too difficult to craft similar items for landscapes, thus clarifying collective goals while placing emphasis on policy.

MHE research reflects and explores these principles [8,110].

## 6. Regional Collaborations

How has this integrated, regional-to-global approach to marine ecosystems been successful in reaching policy makers? For MHE, a variety of international and regional organizations and collaborative communities support the integration and dissemination of research. In the last quarter century, MHE has become a distinct discipline in the marine sciences, bringing a systematic, long-term perspective to the study of interactions between human societies and the world's seas, oceans, and coastlines [109].

Previous research at the global scale offered a valuable platform. The International Geosphere-Biosphere Programme [111] (1987–2015) led a global effort to study how planet Earth functions and how it has—and will—change. A sort of mega-ecotonal framework linked land, sea, and sky projects where they meet: land–atmosphere, land–ocean, and atmosphere–ocean. Building on the IGBP research into planetary biogeophysics, today's

international projects recognize the ubiquitous role of humans and have added contemporary concern for the inclusion people who are affected by decisions made at other scales. To better understand past and present management strategies, today's regional coalitions welcome collaboration with communities of heritage (e.g., indigenous groups) and practice (e.g., fisheries large and small) along with practitioners of archaeology, history, and the social sciences.

Future Earth [112] inherited Land–Ocean Interactions in the Coastal Zone (LOICZ) from IGBP and the International Human Dimensions Programme. Now termed Future Earth Coasts [113], it is a community of several organizations and a platform for translating sustainability knowledge into action. In the spirit of historical ecology, Future Earth Coasts focuses on equitable and sustainable coastal development, community adaptive capacity, and the translation of knowledge into action. Another inherited program from IGBP is the Integrated Marine Biosphere Research (IMBeR) [114]. The IMBeR program supports research to understand, quantify, and compare the historic and present ocean/human systems to predict changes and develop scenarios and options for transitioning towards ocean sustainability.

Dynamic researcher-led programs have also been important. An example is the Oceans Past Initiative [115], which facilitates historical research in marine sciences, policy making, and resource management. OPI addresses the issue of shifting baselines for numerous species and ecosystems, coordinates resources, and provides useful information to the marine historical research community, decision-makers, and the public. A recent European Research Council [61] grant has funded NorFish, an OPI project with focus on the 16th century North Atlantic Fish Revolution as an early example of the disrupting effects of globalization and climate change. In the Pacific northwest, U.S. and Canadian funders support the Clam Garden Network [116] and the Herring School [117]; these collaborations among scholars and indigenous groups cross national boundaries to learn about and rehabilitate ancient marine gardens.

Early and enduring support for Arctic research from the United States' National Science Foundation [118] enabled the researcher-led North Atlantic Biocultural Organization [119] to begin research in 1992. This large collaborative network of scholars and projects is one of the earliest efforts to cross-cut national and disciplinary boundaries and to help North Atlantic scholars to realize the immense research potential of the region. NABO works to improve basic data comparability, to engage in practical fieldwork with indigenous and local participants, to train students and citizen scientists, and to communicate findings to scholars, funding agencies, and the public. NABO also collaborates with the Humanities for Environment Observatories [120], a global network to identify, explore, and demonstrate the contributions that humanistic and artistic disciplines make to solving global social and environmental challenges. These researcher-led programs are largely open for participation to anyone who registers and follows the rules.

As with landscapes, attention to all scales (from local to global) is essential for MHE research, but to reach policy makers the marine researchers have tied together temporally and spatially diverse projects with themes [109]. Some examples are species with both economic and cultural meaning (e.g., herring, salmon), regions or ecosystems at risk (islands, wetlands), or the comparison of community-led and industrial management strategies. This meta-language is intellectually fruitful for researchers and is much more understandable to policy makers.

## 7. Conclusions

What strategies could increase the importance of landscapes and local/regional research to policy makers? Given the diversity of climatic conditions and global ecosystems, no uniform scheme can be sufficient. The place-based, human scale approach of historical ecology, paired with its flexible toolbox, can increase support from planners and administrators, elected officials, citizen-powered associations, and the public.

The need to anchor the elements that comprise landscapes within broader physical and social contexts is a key feature of historical ecology. Considerable landscape research is already addressing many topics of enduring interest to managers. We have seen projects in Sicily and France that anticipate climate change, and additional examples address the global comparison of innovation in agricultural economies [121,122]; the genetic history of crops and contemporary food security [123]; and the economic and environmental consequences of agriculture, forestry, and other practices [124,125]. Engelhard and colleagues' six policy themes (climate change, biodiversity conservation, habitat integrity, ecosystem structure and function, food security, socioeconomic considerations, and more democratic governance) offer some good ideas for increasing this trend and adapting them to landscapes. By consolidating the interests of multiple local and regional stakeholders, thematic research can catch the interest of funding entities and policy makers.

Another important theme of historical ecology is the investigation of baselines. With shared research and clear policy goals, landscape ecology/archaeology can routinely extend baselines, examine contexts, and compare management and policy strategies [90]. This is particularly useful for conservation planning, to evaluate species health, abundance, and social/economic contexts under different management regimes. Species that evoke contentious community response (e.g., wolves, boars, deer) have long been of interest to environmental historians, geographers, and cultural anthropologists whose work links well with zooarchaeology and genomics. Engelhard and colleagues' [109] six policy themes (climate change, biodiversity conservation, habitat integrity, ecosystem structure and function, food security, socioeconomic considerations, and more democratic governance) offer some good ideas for increasing this trend and adapting them to landscapes.

Contentious issues such as these need not end badly; there are means to adjudicate disputes [126–128]. The Harvard University Law School's Program on Negotiation publishes the Negotiation Journal and other teaching materials [129]. Trade-offs can be examined against payoffs [130,131] and ways forward can be found. While environmental regulation is often controversial, it is imperative that solid science and stakeholder voices be a critical component of good policy.

Wherever they work, historical ecological researchers have concentrated on supporting communities that, together with their scholarly collaborators, set goals. These alliances include citizen scientists (especially young people and knowledgeable elders) whose local knowledge enhances discovery, monitoring, and other activities. Collaborative heritage management benefits diverse actors.

Both marine historical ecology and landscape ecology/archaeology communities have found allies in like-minded funding agencies and alliances with international and global entities. Such alliances can offer many forms of support (e.g., funding, publicity, logistics, advice, access). While the MHE researchers were building networks and exploring citizen science, landscape researchers have focused on management technologies and training the next generation of scholars. Of course, the MHE projects were also training scholars, and the landscape projects were reaching out to communities; both have, quite rightly, been fine-tuning important arenas of competence and engagement with funders.

Taken together, they are building an increasingly integrated historical ecology. A research design that uses a complex systems approach (that is, to identify multiple temporal and spatial drivers of systemic change) integrates researchers across disciplinary boundaries and encourages the participation of other stakeholders. However, if all this work is meant to benefit the future, it is now time to wade into areas of conflict, not as proponents of one side or another but to offer peer-reviewed, stakeholder-supported assessments and management options for landscapes and waterscapes across time and space. This is the power of the past for the future.

**Funding:** This research received no external funding.

**Institutional Review Board Statement:** Not applicable.

**Informed Consent Statement:** Not applicable.

**Data Availability Statement:** Not applicable.

**Conflicts of Interest:** The author has no conflict of interest.

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
