# Peer review of "Historical Ecology: A Robust Bridge between Archaeology and Ecology"

_sustainability, doi:10.3390/su13158210_

Round 1

Reviewer 1 Report

This paper is a much welcomed contribution showing the importance of (1) bridging the past with the present (and future) to understand current ecological problems, and (2) developing solutions and strategies based on such an understanding, integrating research results into policies. In this respect, the article presents 'historical ecology' as a framework to study ecosystems, landscapes, and waterscapes in a long-term perspective. One of the main strengths is that the author, an authority in this field, defines directly at the beginning a list of actions for scholars: "increase the pool of expertise by engaging all relevant knowledge communities, collect rapidly disappearing data, analyze with both familiar and new methods, and apply the results of actionable science to policy and practice." My suggestions refer in particular to this action list and the structure of the paper, as I believe that some of the main arguments could be formulated even stronger.

  1. Regarding the first point of the action list, "increase the pool of expertise by engaging all relevant knowledge communities," there is still a lot of potential here in placing greater emphasis on community-based participatory research (CBPR), involving indigenous and local communities. Aspects of CBPR or management practices of local, indigenous, and disenfranchised (LID) communities are mentioned throughout the text, but - given its significance in the context of sustainability - this should be at the forefront of the article and clearly stated in the list of actions for example.
  2. An important point is made in applying "the results of actionable science to policy and practice." It remains unclear, however, of how to actually approach and communicate with policy makers, which certainly seems to be a complex task. I propose to include a (short) section listing best practicable ways of approaching and collaborating with policy makers.
  3. Similarly, I propose to include a section listing the precise methods and techniques of historical ecology, especially to counterbalance the narrative aspect, often floating over many examples at a high level. A "toolbox" of historical ecology is mentioned several times, but never really defined.
  4. The article is managed without any figures and tables, which is generally fine. However, it would be easier for the reader to access the content if there were at least some accompanying materials, such as a graphical scheme of the action list.

Author Response

Thank you for your review.  Here are my responses:

  1. approaching policy makers: This is not within my specialty and I would be guessing ho that is done in MHE circles, and I do not currently know of a paper that discusses this issue.
  2. precise methods & techniques: In my opinion, a discussion of the 'toolbox' is not useful, because as I mention in the article the 'tools' vary from one place to another.  What I have outlined, instead, are the basic elements (community involvement and action, broad range of knowledge communities, etc).
  3. no figures/tables: I could concoct something but I think it unnecessary.

Reviewer 2 Report

Excellent overview of a complex and rapidly evolving area of collaboration and action.  Well written call to action as well as state of field survey, great references.

Author Response

Thank you for your encouraging review.

Reviewer 3 Report

This is a fine article that will assist in unifying the other more regional examples of historical ecology and the role of archaeology for this special issue of Sustainability.  Carole Crumley must be viewed as one of the most, if not the most, influential voice in introducing and sustaining the role of historical ecology for the discipline of archaeology from the outset of this alliance many years ago.  Having her assessment of this intellectual and practical merger by way of a review of where this approach presently stands is surely timely and influential.  She provides background to the development of “a robust bridge between archaeology and ecology” emphasizing the “big tent” of disciplinary inclusiveness as well as the practical role of local and indigenous participants which now identifies historical ecology.  By outlining the several programs, projects and groups now drawing on the approach, she provides the reader a context for the impact it is now having in addressing our changing environmental and societal apprehensions at scalar and spatial levels of assessment-- local, regional and global.  The piece is an evocative call for action today drawing on the human collective and our planetary past.  It is a practical and informative view about the role of research, scholarship and actionable applications for policy decisions, decisions that are immediately necessary if we are to mitigate, adapt and sustain a future relationship with our earth.

Author Response

Thank you for your encouraging review.